# Second Generation Long-Acting Injectable Antipsychotics in Schizophrenia: The Patient’s Subjective Quality of Life, Well-Being, and Satisfaction

**DOI:** 10.3390/jcm12226985

**Published:** 2023-11-08

**Authors:** Claudio Brasso, Silvio Bellino, Paola Bozzatello, Cristiana Montemagni, Marco Giuseppe Alberto Nobili, Rodolfo Sgro, Paola Rocca

**Affiliations:** Department of Neuroscience “Rita Levi Montalcini”, University of Turin, Via Cherasco, 13, 10126 Turin, Italy; silvio.bellino@unito.it (S.B.); paola.bozzatello@unito.it (P.B.); cristiana.montemagni@unito.it (C.M.); marcogiuseppealberto.nobili@unito.it (M.G.A.N.); rodolfo.sgro@unito.it (R.S.); paola.rocca@unito.it (P.R.)

**Keywords:** rapid review, psychosis, adherence, personal recovery, LAI, risperidone, paliperidone palmitate, aripiprazole, olanzapine

## Abstract

Schizophrenia (SZ) is among the twenty most disabling diseases worldwide. Subjective quality of life, well-being, and satisfaction are core elements to achieving personal recovery from the disorder. Long-acting injectable second-generation antipsychotics (SGA-LAIs) represent a valid therapeutic option for the treatment of SZ as they guarantee good efficacy and adherence to treatment. The aim of this rapid review is to summarize the evidence on the efficacy of SGA-LAIs in improving subjective quality of life, well-being, and satisfaction. The PubMed database was searched for original studies using SGA, LAI, risperidone, paliperidone, aripiprazole, olanzapine, SZ, and psychosis as keywords. Twenty-one studies were included: 13 clinical trials, 7 observational studies, and 1 post hoc analysis. It has been shown that SGA-LAIs bring an improvement to specific domains of subjective and self-rated quality of life, well-being, or satisfaction in prospective observational studies without a control arm and in randomized controlled trials versus placebo. The superiority of SGA-LAIs as compared with oral equivalents and haloperidol-LAI has been reported by some randomized controlled and observational studies. Although promising, the evidence is still limited because of the lack of studies and several methodological issues concerning the choice of the sample, the evaluation of the outcome variables, and the study design. New methodologically sound studies are needed.

## 1. Introduction

Schizophrenia (SZ) is a mental disorder that ranks among the 20 top causes of disability according to the World Health Organization and affects about 1% of the population [1]. Patients with SZ may experience positive, negative, affective, and cognitive symptoms, which severely impair daily functioning and require lifelong treatment [2,3,4]. The effective management of SZ requires early intervention and continuous long-term treatment to reduce symptoms, prevent relapse, maintain function, and improve quality of life [5,6,7,8,9]. Antipsychotic drugs are the mainstay of its treatment, however, about 26.5% to 58.8% of people with SZ are non-adherent to their antipsychotic medication regimen [10,11]. Some factors linked with nonadherence include not being well informed about medications and the negative perceptions of taking medications [6,11,12,13,14]. Poor adherence, in turn, is associated with a higher risk of symptom recurrence, rehospitalization, poor quality of life, and healthcare costs [15,16]. Thus, it is essential to understand which modifiable factors affect medication adherence among people living with SZ to act on them.

First, a key factor in improving medication adherence is proper treatment prescription. Two types of antipsychotic formulations are currently available for the treatment of SZ, including long-acting injectables and orals. Compared to daily oral antipsychotics, LAI antipsychotics (LAIs) may facilitate therapeutic continuity and adherence thanks to their peculiar pharmacokinetics, lower dosing frequency, and administration by healthcare providers [2,17,18,19,20]. Current guidelines primarily recommend initiating LAIs among patients with a history of poor or uncertain adherence [21], although the use of LAIs is also recommended as maintenance therapy or in response to patient preference [2,21]. Some guidelines also recommend the use of LAI for the treatment of first-episode schizophrenia [2,19,21,22,23]. LAIs have demonstrated comparable effectiveness to oral antipsychotics but exhibit significantly lower relapse and treatment discontinuation rates and more frequent remissions [24,25,26,27].

Second, subjective satisfaction, well-being, and quality of life clearly support adherence in patients undergoing antipsychotic maintenance treatment [28,29,30,31]. Satisfaction is considered a clinical index of treatment adherence, efficacy, and success and, similarly to adherence, is influenced by treatment response, patients’ attitudes towards drugs, demographic features, cultural background, previous treatment experience, involvement in treatment planning, and by the quality of mental health services and care [28,30,32,33,34,35]. Subjective quality of life generally refers to an individual’s perception of their position in life within the context of their culture and value systems, considering their goals, expectations, standards, and concerns [36,37]. In SZ a lower subjective quality of life was associated with more severe psychiatric symptoms, medication-induced side effects, poor nutrition, reduced physical activity, metabolic syndrome, social isolation, lack of access to environmental resources, stigma, discrimination, cognitive impairment, and limited employment opportunities [38,39,40,41,42,43,44,45]. Subjective well-being represents a core dimension of life and is based largely upon private internal psychological processes related to physical and mental health, and to the values and goals of an individual [46,47,48]. The study and promotion of subjective well-being in people with SZ incorporate patients’ subjectivity in the treatment process and support a holistic approach to tackling the disorder [48].

Apart from their positive relationship with adherence to treatments, these three subjective constructs, i.e., satisfaction, well-being, and quality of life, are often impaired in people living with SZ and they are, per se, a central therapeutic goal [49,50,51,52]. Indeed, they represent a target to achieve personal recovery, which has both an objective domain that is determined by the severity of the symptoms and levels of functioning, and a subjective domain, which is described through various dimensions, including quality of life, well-being, and satisfaction [53,54]. Compared to first-generation, second-generation antipsychotics (SGAs) are associated with a significant improvement in patient-reported quality of life [55]. At the same time, thanks to a lower administration frequency as opposed to daily administered oral drugs, LAIs are expected to improve medication convenience and therefore patients’ satisfaction [28]. Therefore, SGA-LAIs seem to put together the advantages of SGAs with a long-acting formulation and might represent a good therapeutic option to facilitate the achievement of subjective recovery among people living with SZ [28,56].

Only a few studies have compared the effect of SGA-LAI and oral antipsychotic therapies on subjective recovery from SZ and the summary of the scientific evidence on this specific topic is limited. Actually, only three reviews have mainly focused on the satisfaction, quality of life, and subjective well-being of patients in therapy with SGA-LAIs [56,57,58]. Swainston, Harrison and Goa (2004) [58] conducted a systematic review in which they found that, compared to the placebo, risperidone LAI significantly improved health-related quality of life domains and satisfaction. Kisley et al. (2015) [57] performed a systematic search of the randomized controlled trials (RCTs) comparing the frequency of depot administration (two- vs. four-weekly) for equivalent doses of antipsychotics and found no difference in terms of subjective quality of life. Finally, Rocca et al. (2016) [56] conducted a review of the existing literature on the impact of SGA-LAIs on patients’ functioning and quality of life, suggesting that SGA-LAIs are more effective than placebo.

Our research aims at placing significant emphasis on the subjective dimension of recovery in terms of self-reported well-being, satisfaction, and quality of life of patients with SZ treated with SGA-LAIs. We have observed that existing reviews did not primarily target these specific aspects, prompting us to dedicate our focus to these critical dimensions. This approach will contribute to a more complete and updated synthesis of how SGA-LAI treatments can influence subjective recovery in patients with SZ.

## 2. Materials and Methods

A rapid review simplifies the process of conducting a traditional systematic review by streamlining different methods. It aims at efficiently producing evidence for recipients in a cost-effective way, enhancing the overall execution of knowledge synthesis [59,60]. Psychiatrists providing treatment with SGA-LAIs and researchers interested in the quality of life, well-being, and satisfaction of patients receiving these treatments are the intended readers of this rapid review. The decision to use this review approach is based on factors related to the existing guidelines [59,61,62,63]. Those factors are:the constraints of rapid review methods (e.g., limited search) will provide sufficient information and be credible for end-users;the review has a narrow, well-defined scope (e.g., limited population, specific kind of drugs);the amount of evidence on the chosen topic is small;the evidence to summarize is limited in terms of years of interest;the outcome is relevant to clinicians and patients.

### 2.1. Setting the Research Question and Eligibility Criteria

The objective of the study was to review the literature on the subjective well-being, quality of life, and satisfaction of patients undergoing SGA-LAI treatment. To formulate the research question, we utilized the Patient population Intervention Comparator Outcome Timing Setting (PICOTS) framework, which considers the patient population, intervention, comparator, outcome, timing, and setting [64,65,66,67,68,69]. We chose the following PICOTS: adult people with schizophrenia, treatment with SGA-LAIs, placebo, or other antipsychotics. We restricted the language of the published articles to English.

### 2.2. Search Terms and Electronic Searches

We searched PubMed database using the following search strings: ((aripiprazole[Title/Abstract]) OR (olanzapine[Title/Abstract]) OR (risperidone[Title/Abstract]) OR (paliperidone[Title/Abstract])) AND ((quality of life[Title/Abstract]) OR (satisfaction[Title/Abstract]) OR (well-being[Title/Abstract]) OR (subjective[Title/Abstract]) OR (experience[Title/Abstract]) OR (attitude*[Title/Abstract])) AND ((LAI[Title/Abstract]) OR (long-acting[Title/Abstract])) with a date limit of 14 June 2023. All kinds of articles were included in the search and submitted to retrieval.

### 2.3. Screening and Selection Process

Following the Cochrane evidence-informed guidance on conducting rapid reviews [59], we first screened the title and abstract of the records. We excluded articles not relevant to our review, i.e., articles written in languages other than English, on drugs other than SGA-LAIs, or without subjective and self-rated quality of life, well-being, or satisfaction evaluation among outcomes.

The screening and selection process is summarized in Figure 1.

### 2.4. Data Extraction

C.B. prepared a form to define the data to extract. M.G.A.N. extracted data, and R.S. checked for the correctness and completeness of extracted data. We limited extracted data using the included systematic reviews and focusing exclusively on the PICOTS of the research question [59].

### 2.5. Risk of Bias Assessment

In accordance with the existing guidelines on conducting rapid reviews, we evaluated the risk of bias by considering the study design and the appropriateness of the analyses conducted for the outcomes that concerned patients’ satisfaction, subjective well-being, and quality of life among individuals receiving SGA-LAI treatments [59,61,62,63]. In particular, we followed the recent methodological guidance by the Cochrane Rapid Reviews Methods Group [70]. R.S. performed the RoB assessment and C.B. verified the judgements. The tools employed were the Cochrane RoB 2.0 [71] for the randomized controlled trials (RCTs) and the ROBINS-I [72] for the non-randomized studies of interventions (NRSI). The risk grading of the RoB 2.0 is low risk, with some concerns, and high risk. That of ROBINS-I is low, moderate, serious, and critical. Cross-sectional studies were not assessed with a specific tool and were rated exclusively as at high or critical risk of bias. Together, we considered the results of the RoB assessment dividing the studies into four groups: low, moderate, high, and critical risk of bias. Studies with a critical risk of bias were excluded. The procedure for the RoB assessment is represented in Figure 2. The results of the RoB assessment of each study are presented in Figure 3.

### 2.6. Synthesis and Discussion

We conducted a narrative knowledge synthesis in terms of a descriptive summary of the studies included, followed by a discussion on the differences among studies in their PICOTS elements and experimental design [59,61,62,63]. Then, we highlighted the potential limitations arising from the available literature included [59,61,62,63].

## 3. Results

For an easier understanding, before explaining the results of the present review, we describe the scales most often employed by the studies included in this work to assess patients’ subjective quality of life, well-being, and satisfaction.

### 3.1. Scales for the Assessment of General and Health-Related Quality of Life

#### 3.1.1. The World Health Organization Quality of Life Assessment Brief Form—WHOQoL-BREF

The World Health Organization Quality of Life Assessment WHOQoL is a multidimensional scale designed as a self-rated questionnaire to assess quality of life in a wide range of psychological and physical conditions [92,93]. Initially, the WHOQoL Group developed a 100-question form that covered 24 aspects of quality of life comprehensively (WHOQoL-100). However, due to the long nature of the questionnaire, it became challenging for researchers to focus solely on health-related quality of life (HRQoL). As a result, a shorter version called the WHOQoL-BREF was created by selecting two questions from the total health and general QoL aspects, and one question from each of the remaining 24 aspects [94]. The WHOQoL-BREF includes four domains: physical health (PH; 7 items), psychological well-being (PS; 6 items), social relationships (SR; 3 items), and environment health (EH; 8 items). Each question is rated on a 5-point Likert scale, and scores in a range from 1 to 5. Raw scores in each domain are transformed to a 4–20 score based on guidelines, and then linearly converted to a scale from 0 to 100, where a score of “100” represents the highest possible quality of life [94].

#### 3.1.2. 36-Item Short Form Health Survey Questionnaire—SF-36

The 36-item Short Form Health Survey Questionnaire (SF-36) is widely utilized to assess Health-Related Quality of Life [37,95]. It is a self-rated questionnaire. The SF-36 comprises eight scales: physical functioning (PF), physical role (RP), bodily pain (BP), general health (GH), vitality (VT), social functioning (SF), emotional role (RE), and mental health (MH) [95]. Component analyses have revealed that the SF-36 measures two distinct concepts: a physical dimension, represented by the physical component summary (PCS), and a mental dimension, represented by the mental component summary (MCS) [95]. Each scale contributes differently to the calculation of both PCS and MCS measures.

#### 3.1.3. EuroQoL—EQ

The EuroQoL is a generic health-related quality-of-life instrument validated in a large sample of patients with SZ [96]. It consists of three sections. The first section, called the EQ five dimensions (EQ-5D) describes five health-related domains with five items: mobility, self-care, usual activities, pain/discomfort, and anxiety/depression with three levels of severity: ‘no problem’ = 1, ‘some/moderate problems’ = 2, and ‘many problems’ = 3. The patient chooses between the three options according to how she/he feels on the day of the examination. The EQ-5D section 1 generates 243 different health states from perfect health ‘11111’ to many problems in all five dimensions ‘33333’. The second section of the EQ is called the EQ visual analog system (EQ-VAS). The EQ-VAS consists of a 20 cm vertical scale representing a thermometer, with endpoints of the ‘worst’ and ‘best’ imaginable health status (scored 0 and 100, respectively) [65,96]. The patient chooses on the scale according to how he/she feels on the day of the examination. The third part is the EQ-Index (EQ-I), which gives a score to the health status coded in section 1 (EQ-5D) according to a normative population, e.g., the Spanish general population [96]. It usually ranges from perfect health scored 1 to death scored 0 [96].

#### 3.1.4. Schizophrenia Quality of Life Scale—S-QoL

The Schizophrenia Quality of Life Scale (S-QoL) evaluates health-related quality of life (HR-QoL) defined as the discrepancies perceived by patients between their expectations and their current life experiences [97]. It was specifically designed for patients suffering from SZ and covers the domains of HR-QoL that differ from the areas evaluated with other tools and is an efficient instrument for the measurement of the impact of SZ on individuals’ lives [97]. It is composed of 41 items that express a positive situation, condition, or behavior. Each item was accompanied by a 5-point Likert scale, ranging from 1 = ‘less than expected’ to 5 = ‘more than expected’ [97].

### 3.2. Scale for the Assessment of Well-Being

#### 3.2.1. The WHO-5 Well-Being Index (WHO-5)

The WHO-5 well-being index [98] is a short, self-administered measurement of well-being over the last two weeks. It consists of five positively worded items about psychological well-being that are rated on a 6-point scale, ranging from 0 (no time) to 5 (all the time). The raw scores are transformed to a score from 0 to 100, with lower scores indicating worse well-being [98].

#### 3.2.2. The Short form of the Subjective Well-Being under Neuroleptics Scale—SWNS

The Short Form of the Subjective Well-Being under Neuroleptics Scale (SWNS) is a self-report tool used to comprehensively evaluate the effectiveness and quality of drug treatment in schizophrenia, as well as measure patients’ subjective well-being [99]. One notable feature of this scale is its ability to assess patients’ subjective thoughts and feelings independently of their psychiatric disorder. The SWNS is widely used in studies evaluating patients’ quality of life, responses to antipsychotic treatment, and drug side [99].

### 3.3. Scales for the Assessment of Satisfaction with Life and Medication

#### 3.3.1. Satisfaction With Life Scale—SWLS

The Satisfaction With Life Scale (SWLS) is a widely used psychological assessment tool that measures an individual’s overall satisfaction with their life [100]. It can provide valuable information about an individual’s overall sense of well-being, their perception of life’s quality, and their level of contentment with different aspects of life.

The SWLS consists of a short questionnaire with a set of statements about life satisfaction. Respondents are asked to rate their agreement with each statement using a scale. The scale ranges from 1 to 7, with 1 indicating “strongly disagree” and 7 indicating “strongly agree” [100].

#### 3.3.2. Treatment Satisfaction Questionnaire for Medication—TSQM

The Treatment Satisfaction Questionnaire for Medication (TSQM) includes several aspects of treatment satisfaction, such as effectiveness, side effects, convenience, and overall satisfaction [101]. The TSQM is frequently employed in clinical trials, research studies, and healthcare contexts to assess patients’ perspectives and encounters with specific medications.

The TSQM comprises four subscales, each targeting a specific aspect of treatment satisfaction: (i) effectiveness, which evaluates the subjective efficacy of the medication and its impact on the patient’s symptoms or condition; (ii) a side effects subscale that examines side effects or adverse reactions from the medication; (iii) convenience, which assesses factors related to the use and administration of the medication, such as dosing frequency, route of administration, and packaging; and (iv) global satisfaction, which provides an overall assessment of the patient’s satisfaction with the medication, considering all aspects of treatment. The rating scale ranges from 1 to 5 or from 0 to 100 [101].

Characteristics of the scales described above are summarized in Table 1.

One hundred and ninety-five records were obtained from the search on PubMed. Following the algorithm described above and reported in Figure 1, 21 records were included in the review. In detail, 13 clinical trials (four double-blinded randomized clinical trials; two open-label, randomized, controlled trials; one open-label, non-randomized, controlled study; six non-randomized, single-arm clinical trials), 1 observational prospective case–control study, 5 observational cohort studies, 1 cross-sectional study, and 1 post hoc analysis.

For a clearer description, we decided to group the results based on the antipsychotic and the study design. First, we will discuss a study about the overall effect of LAI antipsychotics, without focusing on the evidence on a singular drug, then we will summarize the effect of risperidone LAI (RLAI) in single-arm studies versus placebo in randomized clinical trials (RCTs) versus other antipsychotics in RCTs and in observational studies; the effect of olanzapine LAI (OLAI) versus placebo in one RCT; the effect of aripiprazole LAI (aripiprazole one month—AOM) in single-arm studies versus other antipsychotics in RCTs and observational studies; the effect of paliperidone LAI (paliperidone palmitate—PP) versus other antipsychotics in RCTs and observational studies versus oral paliperidone in RCTs; and finally PP administered every three months (PP3M) vs. paliperidone LAI administered every month (PP1M) in one observational study.

### 3.4. LAIs Versus Oral Antipsychotics in an Observational Case-Control Study

In this section, we included the only study selected in which the effects of different LAIs have been evaluated in comparison to oral antipsychotics without a subdivision based on the specific drug, but rather with the aim of considering the overall differences and similarities between these two treatment categories [29].

Pietrini et al., 2016 [29] published a 6-month, prospective, longitudinal, open-label, non-randomized, case–control, observational study where the authors assessed the difference in subjective well-being and satisfaction between 20 patients treated with oral olanzapine (10–15 mg/day) or paliperidone (9–12 mg/day) and 20 patients switched from oral therapy to LAI olanzapine pamoate (300–405 mg/month) and LAI paliperidone palmitate (100–150 mg/month). It comprised a baseline visit (T0) and a 6-month follow-up visit (T1). Patient-reported outcomes were evaluated at both visits using the SWNS and the SF-36. Between T0 and T1, the LAI-antipsychotic maintenance treatment group demonstrated greater improvement than the oral-antipsychotic maintenance treatment group in the Positive and Negative Syndrome Scale total score (PANSS), in the Drug Attitude Inventory 10 (DAI-10), a 10-item questionnaire to assess the patient’s attitude toward their antipsychotic medication, and in all SWNS dimensions except for social integration. Over 6 months, the LAI-antipsychotic maintenance treatment group experienced an overall enhancement in health-related quality of life (assessed using the SF-36) and improved functioning across various aspects of daily life (assessed using the SF-36). Conversely, the oral- antipsychotic maintenance treatment group reported a notable decline in SF-36 specifically regarding emotional role and social functioning during the same timeframe. The overall risk of bias was rated as moderate because of bias due to confounding, the selection of participants, deviations from the intended interventions, and the measurement of outcomes.

The study is summarized in Table 2.

### 3.5. Risperidone Long-Acting Injectable (RLAI)

#### 3.5.1. RLAI in Single-Arm Studies

A non-randomized, single-arm study carried out at 324 centers in 22 European countries by Möller et al. (2005) [73] investigated the efficacy and tolerability of a direct transition from other antipsychotics to RLAI in patients with stable SZ who required a change of treatment for any reason (e.g., lack of efficacy, side-effects, or poor adherence). Health-related quality of life was assessed at 3 and 6 months after the introduction of RLAI using the SF-36, and satisfaction with the treatment was evaluated after 6 months with a 5-point scale ranging from very good to very poor. All subscales of the SF-36 significantly improved after 6 months with clinically significant improvements (i.e., >5 points) for Physical Role, Bodily Pain, General Health, Social Functioning, Emotional Role, and Mental Health. Patient satisfaction with treatment also improved significantly after 6 months, specifically, the proportion of patients who rated their satisfaction as ‘very good’ increased from 6% at baseline to 31%. The overall risk of bias was rated as serious principally because of bias due to confounding and the measurement of outcomes.

Fleischhacker et al., 2005 [74] in a 1-year, open-label, international multicenter trial of RLAI in 615 stable adult patients with SZ, self-rated functioning and well-being were measured every 3 months using the Short Form 36-item questionnaire (SF-36). Significant improvements were found in the mental component, vitality, and social functioning scales. The overall risk of bias was rated as moderate because of bias due to confounding, classification of the interventions, the measurement of outcomes, and selection of the reported results.

Lee et al., 2006 [75] evaluated the efficacy and safety of RLAI for 48 weeks in Korean patients in a non-randomized, open-label, single-center, 48-week study. Each of the participants visited the hospital every 2 weeks, and injections were given at each visit. Complete evaluations were carried out on five occasions (baseline, 12, 24, 36, 48 weeks). The patients included were clinically stable and directly transitioned to RLAI. To assess subjective quality of life and well-being the SF-36 and the SWNS, respectively, were employed. Forty patients were enrolled, and twenty-five patients completed this study. No differences were found in subjective well-being between the baseline and the end of the follow-up (48 weeks). The overall risk of bias was rated as severe principally because of bias due to confounding.

Nick et al., 2006 [76] performed a non-randomized, single-arm, 6-month multicenter European study. The patients’ quality of life was evaluated using the SF-36 survey at the beginning of the trial, after 3 months, and at the end of the study (6 months). To gauge treatment satisfaction, a 5-point Likert scale ranging from “very good” to “very poor” was employed. Most patients (86.7%) initiated their treatment with a dosage of 25 mg/14 days, whereas the remaining participants were administered either 37.5 mg/14 days or 50 mg/14 days. The trial yielded a notable and statistically significant improvement in treatment satisfaction (*p* < 0.001). Specifically, the proportion of patients reporting their satisfaction as “very good” experienced a substantial increase from 4.3% to 37.0%. Although the SF-36 survey revealed certain positive changes in the patients’ quality of life, it is important to note that these improvements did not attain statistical significance. The overall risk of bias was rated as moderate because of bias due to confounding, the selection of participants, measurement of outcomes, and the selection of the reported results.

Lasser et al., 2007 [77] conducted an open-label 50-week trial including young adults (men aged 18–25 years and women aged 18–30 years). Sixty-six patients received at least one injection of RLAI (25 or 50 mg) every two weeks; 64% of the patients completed the 50-week trial. A dose of 25 mg/14 days was received by 23 patients and 50 mg/14 days by 43 patients. The patient-rated quality of life (SF-36 scores) improved and patients’ attitudes toward the medication were positive (DAI scores). The overall risk of bias was rated as moderate because of bias due to confounding, the measurement of outcomes, and the selection of the reported results.

In the study conducted by Niolu et al., 2015 [78] 27 patients with schizophrenia were enrolled and administered RLAI twice a month at a dosage of 25 mg, 37.5 mg, or 50 mg depending on the individual patient’s clinical needs. To assess the subjective well-being during treatment with risperidone, the authors used the SWN scale, while the S-QoL was employed to evaluate health-related quality of life. Scales were assessed monthly for the first 12 months and then every 6 months. Over the course of 30 months, there was a notable and significant increase in both SWN and S-QoL scores. Post hoc tests indicated significant deviations from the SWN baseline values starting from the eighth month and significant deviations from S-QoL baseline values starting from the eighteenth month. Correlation analysis revealed a strong association between the reduction in mean values of the Scale for the Assessment of Positive Symptoms (SAPS) and an increase in mean SWN and S-QoL values during the 30-month period. Similarly, there was an inverse robust correlation between the reduction in mean SANS values and the increase in mean SWN and S-QoL values. The overall risk of bias was rated as moderate because of bias due to confounding and the selection of participants.

#### 3.5.2. Risperidone LAI vs. Placebo in Double-Blind, Randomized, Clinical Trials

Nasrallah et al., 2004 [79] set up a 12-week, multicenter, randomized, double-blind, parallel-group trial to evaluate the efficacy and safety of RLAI (25, 50, or 75 mg, every 2 weeks). Following a one-week evaluation period, patients underwent a gradual adjustment to a 4 mg/day oral dosage of risperidone. They were then randomly assigned to receive either placebo or RLAI every two weeks (options: 25 mg, 50 mg, or 75 mg). During the initial three weeks of the blinded phase, patients in the placebo group received oral placebo, while those assigned to the RLAI group received both oral and injected risperidone according to their prescribed dosage. The SF-36 assessment scale was used to evaluate health-related quality of life. The primary outcome of this study was the change from baseline in SF-36 scores between patients in the risperidone groups and the other patients treated with placebo. By the 12th week, patients who received placebo experienced a decline in health-related quality of life compared to the baseline in seven of the eight domains of the SF-36. Conversely, patients treated with long-acting risperidone showed a significant improvement from the baseline in five domains (of the SF-36 when compared to the placebo (*p* < 0.05). The 25 mg group demonstrated a significant improvement (*p* < 0.05) from the baseline in six of the eight domains of the SF-36. The 50 mg and 75 mg groups exhibited significant improvement in two and three domains, respectively. The overall risk of bias was rated as low. The only concerns were related to the selection of the reported results.

Isitt et al., 2016 [80] analyzed the data derived from an 8-week double-blind, randomized, placebo-controlled, phase 3 study that assessed the efficacy, safety, and tolerability of an R-LAI subcutaneous extended-release formulation (RBP-7000) at 90 mg and 120 mg dosages compared with placebo in subjects with acute SZ (*n* = 337). Health-related quality of life was measured with the EuroQol 5 dimensions index (EQ-I) and visual analog scale (EQ-VAS). Well-being was assessed using the SWNS and satisfaction using the Medication Satisfaction Questionnaire (MSQ), a single-item questionnaire that evaluates satisfaction with antipsychotic medication in SZ ranging from 1 ‘Extremely Dissatisfied’ to 7 ‘Extremely Satisfied’ [102].

The EQ-VAS increased significantly in the risperidone 120 mg group compared to placebo (*p* = 0.0212). In the risperidone 120 mg group, subjects reported significant improvements in SWNS in terms of physical functioning (*p* = 0.0093), social integration (*p* = 0.0368), and total score (*p* = 0.0395). Subjects were significantly more satisfied with risperidone versus placebo (90 mg *p* = 0.0009, 120 mg *p* = 0.0006) and preferred risperidone to their previous medication (90 mg *p* = 0.0001, 120 mg *p* = 0.0619). Significantly, greater improvements in risperidone and overall well-being were demonstrated in patients randomized to long-acting risperidone compared to placebo. The effect was more pronounced in the RBP-7000 120 mg group. The overall risk of bias was rated as low. The only concerns were related to the selection of the reported results.

Litman et al., 2023 [81] published a study consisting of two phases: a 12-week double-blind (DB) phase and a 52-week open-label extension (OLE) phase. In the DB phase, a total of 438 patients experiencing an acute exacerbation of schizophrenia were randomly assigned (1:1:1) to receive either once-monthly intramuscular injections (every 28 days) of RLAI based on in situ microparticles (ISM) at a dosage of 75 mg (*n* = 145), or 100 mg (*n* = 146), or placebo (*n* = 147). Patient-reported subjective well-being under neuroleptic treatment was evaluated using the SWN-S. Assessment scales were administered at multiple time intervals during the study, namely at 4, 8, and 12 weeks in the double-blind phase, and at 12, 24, and 52 weeks in the open-label extension phase. Statistically significant distinctions were observed when comparing the placebo group to the Risperidone ISM group for the “Social integration” domain at Day 29 (*p* = 0.0060) and Day 85 (*p* = 0.0357), as well as for the “Mental functioning” domain at Day 85 (*p* = 0.0184). In the open-label phase, patients were categorized into unstable, stabilized, and stable groups according to their clinical severity, established at the beginning of the OLE phase. Evaluating the SWN-20 total score at weeks 12, 24, and 52 for these individuals, unstable patients showed a noteworthy improvement from baseline (*p* < 0.05 after 12 and 52 weeks; *p* < 0.01 after 24 weeks), stabilized patients demonstrated significant improvement at week 52 (*p* < 0.05), while stable patients maintained consistent scores throughout the follow-up period. We reported some concerns in terms of risk of bias due to the randomization process, deviations from the intended interventions, and the selection of the reported results.

#### 3.5.3. Risperidone LAI vs. Other Antipsychotics in Observational Studies

Mihajlović et al., 2011 [82] conducted a cross-sectional study to evaluate the quality of life of 60 patients affected by schizophrenia in treatment with haloperidol depot or RLAI for one year. Two groups (*n* = 30) were formed based on medication type. The haloperidol depot dosage was 50 mg given every 4 weeks, and RLAI was given in doses of 25 mg and 50 mg every 2 weeks. To measure quality of life, the researchers used two scales: the SWLS and the WHOQoL-BREF. Patients treated with RLAI showed a statistically significant increase in WHOQoL-BREF scale scores compared to those taking haloperidol depot (*p* ≤ 0.05). In particular, they showed a greater inclination to actively engage in everyday activities and leisure. Moreover, the RLAI group achieved better scores with a significant difference (*p* ≤ 0.05) in the SWLS. They had better results in satisfaction with life, living conditions, and self-image than patients treated with haloperidol, with significant variations in satisfaction with health and sleep (*p* = 0.029), the ability to perform daily tasks (*p* = 0.022), and self-fulfillment (*p* = 0.017). Due to the cross-sectional design of the study, we considered it at high risk of bias.

Fe Bravo-Ortiz et al., 2011 [83] conducted an observational study with 1865 patients. The study focused on patients with schizophrenia initiating a new antipsychotic treatment. Baseline assessments gathered sociodemographic data, psychiatric history, and baseline measurements. Prospective follow-up visits at 3 and 6 months assessed disease severity, quality of life, caregiver burden, treatment adherence, and physician satisfaction. The study included three visits: at baseline, 3 months, and 6 months. At the baseline visits the authors collected sociodemographic data, psychiatric history, diagnosis, onset duration, substance abuse, hospitalizations, and reasons for treatment changes. The parameters assessed at baseline and follow-up visits included clinical severity (CGI-S), quality of life (EQ-5D), caregiver burden, PANSS scores, adherence to treatment (subjective assessment by specialist), the satisfaction of the treating physician (subjective assessment on a scale of 0–10 with higher numbers indicating greater satisfaction), adverse events, and the types of antipsychotic medication used. Medication types included injectable/oral conventional/atypical antipsychotics including RLAI. During follow-up, in the RLAI group, a higher percentage of patients showed good EQ-5D scores, as compared to oral first-generation antipsychotics (FGA), oral SGA, and FGA-LAI. The statistical analysis found a significant association (*p* = 0.0018) between antipsychotic type (RLAI, oral FGA, FGA-LAI, and oral SGA) and ED-5D scores. RLAI had an odds ratio of 1.654 (95% CI 1.126–2.431) for achieving a better quality of life, making it favorable compared to the other antipsychotic treatments. The overall risk of bias was rated as moderate because of bias due to confounding, the classification of interventions, the measurement of outcomes, and the selection of the reported results.

The summary of the included studies on RLAI are summarized in Table 3.

### 3.6. Olanzapine LAI (OLAI)

#### Olanzapine LAI vs. Placebo in a Double-Blind, Randomized, Clinical Trial

Witte et al., 2012 [84] in a randomized, double-blind, placebo-controlled trial, compared four treatment arms (OLAI 210 mg/2 weeks; OLAI 300 mg/2 weeks; OLAI 405 mg/4 weeks; and placebo) assessing the mean change from baseline to endpoint (after 8 weeks) of the two component scores and eight subscale scores of the SF-36. The study included 404 patients with SZ with moderate to severe symptoms on the BPRS ≥30 points. The 300 mg/2 weeks and 405 mg/4 weeks OLAI groups and the combined OLAI group were superior to the placebo on the mental component summary score of the SF-36. Each OLAI group and the combined OLAI group were superior on the Mental Health scale of the SF-36. We reported some concerns in terms of the risk of bias due to the randomization process and the measurement of the outcome.

The study is summarized in Table 4.

### 3.7. Aripiprazole LAI Once a Month (AOM)

#### Aripiprazole LAI Once a Month in Single-Arm Studies

In a multicenter, prospective, non-interventional study by Schöttle et al., 2020 [85] 242 symptomatically stable patients with SZ who switched their treatment to AOM after 9.7 (±22.3) months of oral treatment were clinically followed for a 6-month observation period. Their well-being was measured using the WHO-5 index. The study aimed at determining if the AOM treatment in typical care would improve functional status and well-being. Patients were assessed every 4 weeks (baseline assessment + 6 follow-up visits), in which authors administered the WHO-5 index. At the end of the study, well-being significantly improved, with a 4.8-point increase in the WHO-5 index at the endpoint. The progress was most prominent in the first 4 weeks. Out of the participants, 77.9% experienced enhanced well-being, 5.5% remained unchanged, and 16.6% had decreased well-being. Patients ≤35 years old showed a 5.6-point increase, while those over 35 had a smaller increase of 4.4 points. The overall risk of bias was rated as high mainly because of bias due to confounding and the selection of participants.

McEvoy, 2021 [86] performed a post hoc analysis of clinical trial data that evaluated long-term, self-reported mental and physical health-related quality of life assessed using the SF-36 in SZ patients receiving AOM. The study population included 291 stable SZ outpatients enrolled in two consecutive long-term safety studies on AOM given every 4 weeks for up to 124 weeks. The primary outcome was a change in the SF-36 mental component summary (MCS) and physical component summary (PCS) scores from baseline to 124 weeks. The results from this post hoc analysis indicated that the mean MCS score for patients continuing AOM improved significantly from the baseline over 124 weeks (*p* < 0.05, all time points), while the mean PCS score showed little change over the 124 weeks. At baseline, patients had lower (worse) MCS scores than the normed general population, but by week 124, patients had MCS scores comparable to those in the general population. In this post hoc analysis, outpatients with schizophrenia who continued the aripiprazole LAI showed a gradual and sustained improvement in self-reported mental quality of life over several years of follow-up (long-term extension to 180 weeks), whereas the self-reported physical health-related quality of life did not change. The overall risk of bias was rated as moderate because of bias due to confounding, the classification of interventions, the measurement of outcomes, and the selection of the reported results.

The summary of the included studies on AOM are summarized in Table 5.

### 3.8. Paliperidone Palmitate LAI (PP)

Paliperidone palmitate LAI exists in three formulations with different durations: once-monthly paliperidone palmitate (PP1M), trimestral paliperidone palmitate (PP3M), and the recently released biannual paliperidone palmitate (PP6M).

#### 3.8.1. Paliperidone LAI vs. Other Antipsychotics

##### Paliperidone LAI vs. Other Antipsychotics in Open-Label, Randomized Clinical Trials

Kwon et al. (2015) [87] performed a 21-week, multicenter, randomized, open-label comparative study. A total of 154 patients with SZ who were unsatisfied with their current oral atypical antipsychotics were enrolled. Dissatisfaction was defined as a score of 4 (‘neither dissatisfied nor satisfied’) or less measured on the Medication Satisfaction Questionnaire (MSQ) [102]. Participants were randomly assigned to either an immediate or delayed (after the end of the study) switching to PP1M. The MSQ and the Treatment Satisfaction Questionnaire for Medication (TSQM) were used to evaluate patients’ satisfaction with treatment, whereas the Positive and Negative Syndrome Scale (PANSS) and the Personal and Social Performance (PSP) scale were used to evaluate efficacy. From baseline to the final assessment, the MSQ score increased significantly in both groups, and the increase was greatest after the first administration of PP1M in the immediate switch group. The immediate switch group showed a significant improvement in the TSQM convenience score compared with the delayed switch group on oral antipsychotics during the comparison period. Most adverse events were minor and tolerable. In short, switching from oral atypical antipsychotics to PP1M because of poor satisfaction significantly improved patient satisfaction, with comparable efficacy and tolerability. We considered this study to be at high risk of bias mainly because of the deviations from the intended results.

In an open-label, randomized, controlled study, Takekita et al., 2016 [88] evaluated the well-being of 30 patients, comparing patients who continued an RLAI twice-monthly treatment with those who were randomly assigned to PP1M. The change in the SWNS was a secondary outcome. The authors found improvements in SWNS scores from baseline for the group that switched to PP1M, particularly for the Emotional Regulation subscale. The study was classified at a high risk of bias principally because of the deviations from the intended results.

##### Paliperidone LAI vs. Other Antipsychotics in Observational Studies

Sağlam Aykut, 2019 [89] selected 84 patients, 33 of them treated with PP1M and 51 with second-generation oral antipsychotics. Each patient underwent a battery of assessments aimed at evaluating symptom severity (PANSS, CGI), side effects (Extrapyramidal Symptom Rating Scale, ESRS; Ugvalg for Kliniske Undersgelser Side Effect Rating Scale, UKU Side Effect Rating Scale), health-related quality of life (SF-36), medication adherence (the Morisky Medication Adherence Scale, MMAS), and insight (the Schedule for Assessing the Three Components of Insight, SATCI). The prescribed dose intervals for PP1M ranged from 100 to 150 mg per month. In the SF-36, patients taking second-generation oral antipsychotics had lower scores in the general subscale compared to those using PP1M (*p* < 0.001). No significant differences were found in the other subscales. The overall risk of bias was rated as moderate because of bias due to the selection of participants, deviations from intended interventions, the measurement of outcomes, and the selection of the reported results.

Di Lorenzo et al., 2022 [90] in an observational cohort study with 90 patients, compared PP3M, PP1M, and haloperidol decanoate (HLAI) treatments. At 6 and 12 months of treatment, they administered the CGI, GAF, and WHOQoL-BREF, and after 1 year of treatment, they evaluated relapses (psychiatric hospitalizations and urgent consultations), side effects, and drop-outs. There were no statistically significant differences among the three treatments in terms of health-related quality of life. We assessed a low risk of bias.

#### 3.8.2. Paliperidone LAI vs. Oral Paliperidone in an Open-Label, Randomized Clinical Trial

Bozzatello et al., 2018 [28] conducted an open-label, randomized, controlled trial aimed at evaluating the efficacy and tolerability of PP1M compared with oral paliperidone with an extended-release (ER). They focused on the satisfaction, subjective well-being, and service engagement of patients. Seventy-two consecutive outpatients with SZ were randomly assigned for 6 months to PP1M (50–150 mg equivalent) or paliperidone ER (6–12 mg/day). Participants were assessed at baseline and after 6 months using the TSQM, SWNS, the Service Engagement Scale (SES), the Clinical Global Impression–Schizophrenia (CGI–SCH), and the Personal and Social Performance (PSP) score. We considered this study to be at high risk of bias mainly because of the deviations from the intended results.

#### 3.8.3. PP3M vs. PP1M in an Observational Study

Fernandez-Miranda et al., 2021 [91] conducted a prospective, observational, open-label study examining the differences in satisfaction and well-being between the use of PP3M and PP1M, without dose stratification of the sample. The study observed 84 patients for 24 months. Patients were treated with PP3M after 2 years of PP1M stabilization. Treatment satisfaction with PP3M vs. PP1M was evaluated using the TSQM and a Visual Analogue Scale (VAS), ranging from 1, not at all satisfied, to 10, extremely satisfied. TSQM and VAS ratings showed significant improvement following the administration of PP3M (*p* < 0.01 and *p* < 0.001, respectively). Patients reported increased satisfaction attributed to a decrease in the number of injections, reduced sedation, and a diminished perception of being medicated. The overall risk of bias was rated as moderate because of bias due to confounding, the selection of participants, deviations from intended interventions, the measurement of outcomes, and the selection of the reported results.

The summary of the included studies on PP are summarized in Table 6.

## 4. Discussion

Although SGA-LAIs have been demonstrated as useful therapeutic options employed in clinical practice for the treatment of SZ for almost thirty years, the focus on the patients’ subjective perspective on their satisfaction, well-being, and quality of life in relation to the administration of these drugs is partly missing. Indeed, in the present review, we found and included only twenty-one pertinent studies published between 2005 and 2023, compared to the hundreds of studies published on SGA-LAIs in the same period.

### 4.1. Synthesis of the Reviewed Studies

The research reviewed in this work demonstrated an improvement in the specific domains of quality of life, satisfaction, or well-being for all SGA-LAIs available in prospective observational studies without a control arm [59,60,63] and in randomized controlled trials versus placebo [65,66,68]. Some studies evidenced the superiority of SGA-LAIs compared to oral equivalents [19,20,93,94] and haloperidol-LAI [69,95]. Finally, two works made a comparison between SGA-LAIs [94,96]. Fernández-Miranda et al., 2021, compared two different formulations of the same drug, showing a higher improvement in satisfaction with PP3M compared to PP1M [96], and Takekita et al., 2016 did not find any significant difference in the subjective well-being of patients treated with RLAI every 15 days or PP1M [94].

### 4.2. Methodological Concerns about the Included Studies

Despite these promising results, the strength of the evidence of these studies appears to be limited by several factors related to the samples of these studies, their experimental design, the number of studies for each drug and the dosage employed in the clinical practice, and their assessment of the subjective quality of life, satisfaction, and well-being.

#### 4.2.1. Methodological Concerns Related to the Sample Selection

Study populations were heterogeneous between studies, mainly in terms of inclusion criteria. Some included patients during acute phases of the disorder [68,71] and others in periods of clinical stability lasting more than one year [69,96]. Some others required a failure with a previous oral antipsychotic treatment [28] or the need to start a new treatment [70]. Only Kwon et al., 2015 [93] requested patients’ dissatisfaction with the previous oral therapy. Also, the age at the baseline evaluation was significantly variable between studies, ranging from a mean age of 23.3 years [63] to more than 50 years [95]. As to the sample size, it varied from 30 [94] to 1865 patients [70]. This high variability between the characteristics of the patients included and the sometimes-small sample size of some studies might reduce the statistical power and the external validity of the positive results reported. Moreover, non-significant results on this topic may be unavailable due to publication bias.

#### 4.2.2. Methodological Concerns Related to the Study Design

Concerning the design of the studies, we found a high degree of heterogeneity. Some papers were trials or observational studies without a comparator, i.e., placebo or other antipsychotic [59,60,61,62,63,64,92], others were longitudinal studies comparing an SGA-LAI with an oral SGA [19,70,94], with a FGA-LAI [95], with another SGA-LAI [94], or with a different formulation in terms of administration frequency of the same SGA-LAI, i.e., PP1M vs. PP3M [96]. Four studies were RCT confronting SGA-LAI versus placebo [65,66,68,71], while no trial made a direct comparison between two different SGA-LAIs considering the patients’ subjective and self-rated quality of life, well-being, and satisfaction.

Also, the observation period of the studies was highly variable, from the cross-sectional design of Mijailovic et al., 2011 [69] to a 2.5-year follow-up [64]. These differences limit the soundness and the generalizability of the results of the studies.

#### 4.2.3. Methodological Concerns Related to the Low Number of Studies for Each SGA-LAI

The number of studies supporting the positive relationship between the use of SGA-LAIs and the improvement in patients’ subjective well-being, perceived quality of life, and satisfaction is very small when focusing on a single SGA-LAI. In this paper, for example, we found and included 11 articles on RLAI, but only 1 on OLAI. The included studies on AOM and PP are few, namely two and six, respectively. This imbalance in the number of studies for each SGA-LAI might depend on the period since the development and approval of the drugs for the treatment of SZ (RLAI was the first SGA-LAI developed and approved) or, in the case of OLAI, on the greater difficulties related to storage and monitoring after administration of the drug that may limit the use of this specific SGA-LAI due to clinicians or patients’ satisfaction in treatment. In addition, the variability within each SGA-LAI related to available dosages and formulations should be considered. For PP alone, there are currently three LAI formulations namely PP1M, PP3M, and PP6M, each available at different dosages. The situation is similar for OLAI and RLAI available at multiple dosages in formulations to be administered either every 15 days or monthly. Moreover, RLAI administered once a month exists in a subcutaneous (RBP 700) and an intramuscular (ISM) formulation. Given the high variability in dosages and formulations, it would be necessary to conduct more studies aimed at comparing not only different SGA-LAIs but also different dosages and formulations of the same SGA-LAI as Bozzatello et al., 2018 [28] and Fernandez et al., 2021 [96] have conducted for PP.

#### 4.2.4. Methodological Concerns Related to the Outcome Measures

Finally, some methodological concerns are related to the assessment of satisfaction, subjective quality of life, and well-being. Various validated instruments were employed in the included study, however, a gold standard evaluation of the patients’ subjective perspective about antipsychotic treatments is still missing. The absence of common tools again limits the external validity of the results of each study and therefore the between-study comparability.

#### 4.2.5. Possible Methodological Solutions for Further Studies on This Topic

A possible way to overcome these limitations is the constant inclusion of subjective, self-rated quality of life, well-being, and satisfaction among the primary outcomes of studies on SGA-LAIs. In particular, large umbrella trials and long-term, real-world observational longitudinal studies confronting different SGA-LAIs or different dosages and formulations of the same SGA-LAI are needed to confirm the results of this review.

### 4.3. Limitations and Strengths

The main limitations of this work are the rapid nature of the review, which did not allow for accurate screening of the methodological quality of the included records, and the choice to search the PubMed platform exclusively and no other databases such as Web of Science. The main strength is the choice of the topic covered, which has been previously poorly addressed by other reviews. Indeed, this rapid review represents the first synthesis of the evidence on the efficacy of the four SGA-LAIs approved for the treatment of SZ in the improvement of subjective quality of life, well-being, and satisfaction.

## 5. Conclusions

In conclusion, SGA-LAIs represent a valid option for the treatment of SZ that might positively affect subjective well-being, quality of life, and satisfaction. However, the evidence on the efficacy of these drugs is still limited because of the lack of studies on this topic and because of several methodological issues concerning the choice of the sample, the evaluation of the outcome variables, and the study design. Almost all included studies showed an improvement in at least one aspect of subjective well-being and quality of life. Therefore, according to the available evidence synthesized in the current review, SGA-LAIs may represent a valid therapeutic option not only in terms of the treatment of the symptoms of SZ but also for patients’ subjective well-being. New methodologically sound studies confronting oral treatments and SGA-LAIs with a clear focus on subjective quality of life, well-being, and satisfaction are needed to confirm the available evidence on this topic.

## Figures and Tables

**Figure 1 jcm-12-06985-f001:**
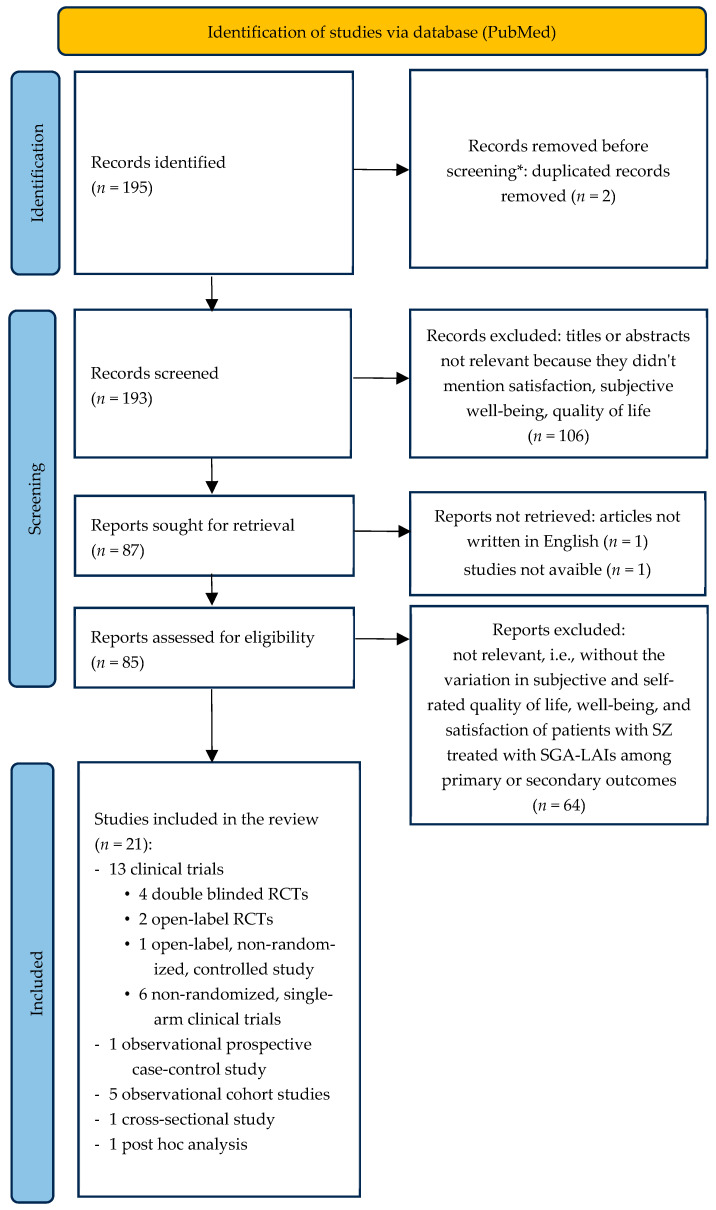
Flowchart of the selection process of the articles included. * No automation tools were used. SGA-LAIs: long-acting injectable second-generation antipsychotics; RCT: randomized clinical trials.

**Figure 2 jcm-12-06985-f002:**
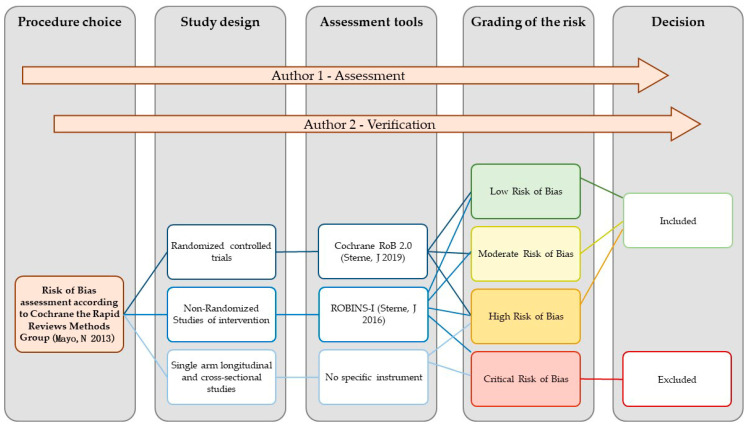
Flowchart of the risk of bias assessment procedure [64,71,72].

**Figure 3 jcm-12-06985-f003:**
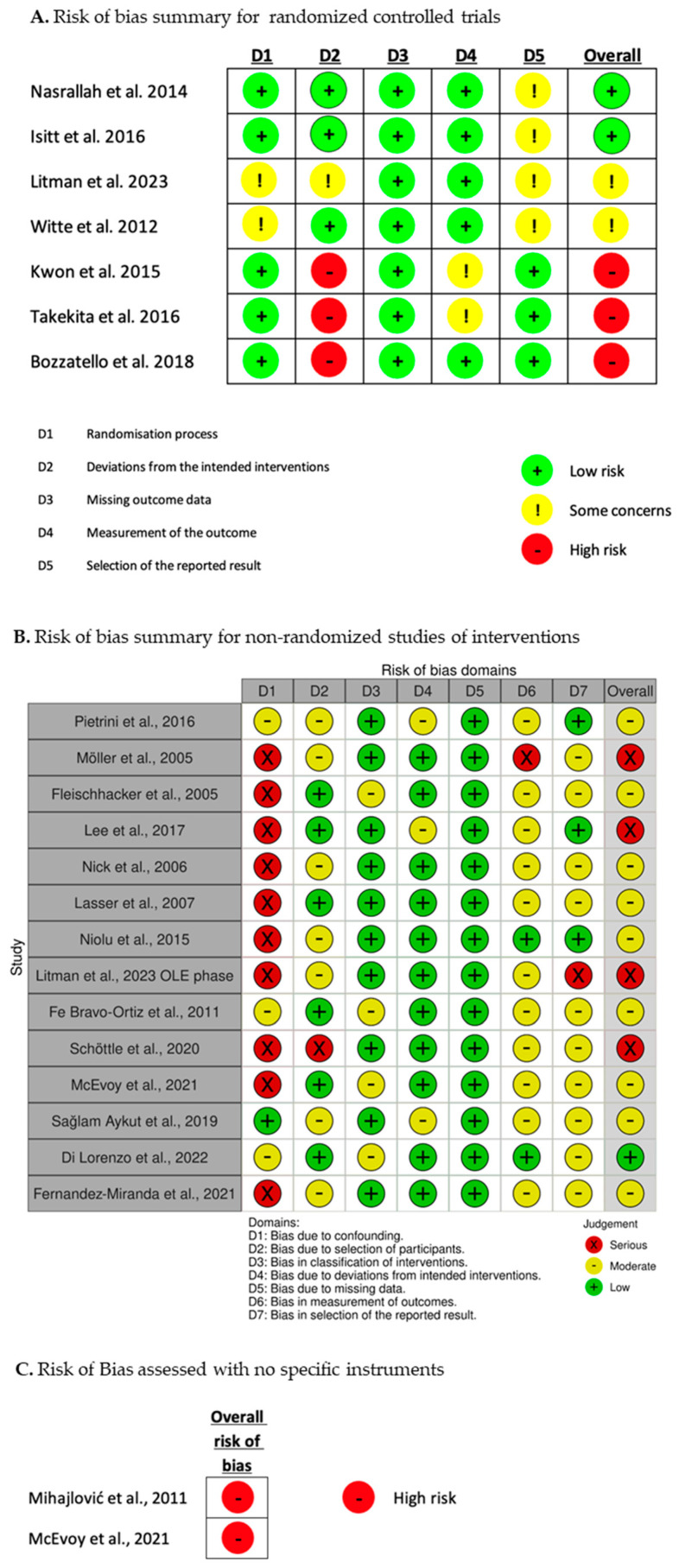
Results of the risk of bias assessment [28,29,73,74,75,76,77,78,79,80,81,82,83,84,85,86,87,88,89,90,91].

**Table 1 jcm-12-06985-t001:** Most employed scales to assess subjective quality of life, well-being, and satisfaction in the included studies.

Acronym	Extended Name	Items Number	Subscales, Components, Items	Single Item Scoring
WHOQoL-BREF WHOQOL group, 1998 [94]	World Health Organization Quality of Life Assessment Brief Form	24	Subscales: physical health (PH; 7 items); psychological well-being (PS; 6 items); social relationships (SR; 3 items); environment health (EH; 8 items)	1 to 5
SF-36 Ware et al., 1993 [95]	36-Item Short Form Health Survey questionnaire	36	Subscales: physical functioning (PF); physical role (RP); bodily pain (BP); general health (GH); vitality (VT); social functioning (SF); emotional role (RE); mental health (MH) Components: physical component summary (PCS); mental component summary (MCS)	Depending on the item
EQ Kind, 1996 [96]	EuroQoL	5	EQ 5 dimensions (EQ-5D): mobility, self-care, usual activities, pain/discomfort, and anxiety/depression; EQ visual analog scale (EQ-VAS); EQ Index (EQ-I)	1 to 3
S-QoL Auquier et al., 2003 [97]	Schizophrenia Quality of Life Scale	41	Subscales: psychological well-being, self-esteem, family relationships, relationship with friends, resilience, physical well-being, autonomy, and sentimental life	1 to 5
WHO-5 Topp et al., 2015 [98]	WHO-5 Well-Being Index	5	Five positive questions about the time spent in a state of psychological well-being	0 to 5
SWNS Naber et al., 1995 [99]	Short Form of the Subjective Well-Being under Neuroleptics Scale	20	Emotional Regulation (ER); Mental Functioning (MF); Physical Functioning (PF); Self-Control (SC); Social Integration (SI)	1 to 6
SWLS Diener et al., 1985 [100]	Satisfaction With Life Scale	5	Items: in most ways, my life is close to my ideal; the conditions of my life are excellent; I am satisfied with my life; so far, I have gotten the important things I want in life; if I could live my life over, I would change almost nothing	1 to 7
TSQM Atkinson et al., 2004 [101]	Treatment Satisfaction Questionnaire for Medication	12	Subscales: effectiveness; side effects; convenience; global satisfaction	1 to 5

**Table 2 jcm-12-06985-t002:** Summary of an observational study on olanzapine and paliperidone LAIs.

Study	Study Design	Drugs	Comparator Group(s)	Sample Size	Age	Sample Characteristics	Diagnosis	Assessment of QoL/SWB/Sat	Outcome	Results	
										Change in QoL/SWB/Sat	Effect
Pietrini et al., 2016 [29]	Observational case–control longitudinal study	LAI group (OLA 300–405 mg/month and PP 100–150 mg/month)	Oral group (OLA 10–15 mg/d or PAL 9–12 mg/d)	40	LAI group: 40.55 (11.00) Oral group: 45.10 (10.68)	Patients in full remission, treated with a single antipsychotic (olanzapine or paliperidone) for over 4 weeks, eligible for a switch to the equivalent LAI maintenance regimen of the same antipsychotic.	SZ	SWNS, SF-36	Primary Outcome: change in SWNS and SF-36 at month 6	↑ vs. oral group in single domains of SF-36 and SWNS	LAI group vs. oral group: SWNS significant improvement in single domains: ER *p* < 0.05 SC, PF, SI *p* < 0.01 MF *p* < 0.001 LAI group vs. baseline: SF-36 significant improvement in single domains: RF *p* < 0.05 SF, VT, RE, MH *p* < 0.01 GH *p* < 0.001 Oral group vs. baseline: SF-36 significant worsening in single domains: SF, RE *p* < 0.05

↑: increase/higher; ER: Emotional Regulation; GH: General Health; LAI: Long-Acting Injectable; MF: Mental Functioning; MH: Mental Health; OLA: Olanzapine; PF: Physical Functioning; PP: paliperidone palmitate; QoL: quality of life; RE: Emotional Role; Sat: satisfaction; SC: Self Control; SI: Social Integration; SF: Social Functioning; SF-36: Short Form-36 Health Survey; SWB: subjective well-being; SWNS: The Short Form of the Subjective Well-being under Neuroleptics scale; SZ: schizophrenia; VT: Vitality.

**Table 3 jcm-12-06985-t003:** Summary of the included studies on risperidone LAI (RLAI).

Study	Study Design	Drugs	Comparator Group(s)	Sample Size	Age	Sample Characteristics	Diagnosis	Assessment of QoL/SWB/Sat	Outcome	Results	
										Change in QoL/SWB/Sat	Effect
Möller et al., 2005 [73]	Non-randomized, single-arm study	RLAI-TM 25–50 mg	N/A	1876	39.8 (12.1)	Symptomatically stable, requiring a therapy change	SZ	SF-36 5-point scale for satisfaction with treatment	Improvements in PANSS, CGI-S, GAF, SF-36, ESRS	↑ SF-36 and Sat with treatment vs. baseline	↑ vs. baseline for all SF-36 domains; > 5 pts difference for RP, BP, GH, SF, RE, and MH domains of SF-36 ↑ Sat. with treatment (*p* < 0.001)
Fleischhacker et al., 2005 [74]	Open-label, single arm, non-controlled multicenter trial	RLAI-TM 25–75 mg	N/A	400	42 (15.0)	Symptomatically stable	SZ	SF-36	Primary: PANSS; Secondary: SF-36	↑ SF-36 vs. baseline	↑ vs. baseline for SF-36 MHC (*p* < 0.08)
Lee et al., 2007 [75]	Non-randomized, open-label, single-center trial	RLAI-TM 25–50 mg	N/A	40	37 (10.5)	Symptomatically stable	SZ, SZA	SF-36 SWNS	Improvements in CGI-S, PANSS, GAF	SF-36 ↔ vs. baseline SWNS ↔ vs. baseline	Patient-rated perceived functioning and well-being did not change from baseline
Nick et al., 2006 [76]	Open-label, single-arm study	RLAI-TM 25–50 mg	N/A	60	40.5 (12.0)	Symptomatically stable and considered to require a treatment change	SZ, SZA	SF-36 5-point scale for satisfaction with treatment	Primary outcome: change in PANSS total score Secondary outcome: change in CGI-S, GAF, SF-36, and treatment satisfaction	↔ vs. baseline for SF-36 ↑ vs. baseline for the 5-point treatment satisfaction questionnaire	Treatment satisfaction improvement (*p* < 0.001). The proportion of patients rating their satisfaction as “very good” increased from 4.3% at baseline to 37.0% at endpoint
Lasser 2007 [77]	Open-label trial	RLAI-TM 25–50 mg	N/A	66	23.3 (3.3)	Men aged 18–25 years and women aged 18–30 years	SZ	SF-36	Improvements in PANSS, CGI–S, SF-36, DAI	↑ of SF-36 vs. baseline	Improvements in all SF subscales except BP; improvements > 5 pts on the SF, RE, PF, and RP subscales
Niolu et al., 2015 [78]	Open-label, single-arm study	RLAI-TM 25–50 mg	N/A	27	36.1 (NA)	Patients with frequent episodes of recurrence with hospital admission because of poor adherence to treatment	SZ, SZA	S-QoL SWN	Change from baseline after 30 months in SANS, SAPS, S-QoL, SWN	↑ vs. baseline for SWN	S-QoL *p* < 0.01 SWN *p* < 0.0001
Nasrallah et al., 2004 [79]	Double-blinded RCT	RLAI-TM 25–75 mg	Placebo	369	25 mg: 38.9 (1.0); 50 mg: 36 (1.0); 75 mg: 39 (1.1) placebo: 38.2 (0.9)	PANSS total score between 60 and 120 pts	SZ	SF-36	Primary outcome: change after 12 weeks in single domains of SF-36	↑ vs. placebo for SF-36	RLAI total group vs. placebo: *p* < 0.05 in SF-36 RS, BP, GH, SF, RE, MH RLAI 50 mg vs. placebo: *p* < 0.05 in SF-36 BP, MH RLAI 75 mg vs. placebo: *p* < 0.05 in SF-36 BP, RS, FE
Isitt et al., 2016 [80]	Randomized, double-blind, placebo-controlled, multicenter phase 3 study	RLAI-OM 90 or 120 mg RBP-700, Sbc	Placebo	337	18–55, median 43	PANSS between 80 and 120 pts	SZ	SWNS MSQ	Primary outcome: change in EQ SWNS MSQ	↑ vs. placebo for SWNS, EQ-VAS, MSQ	RBP-7000 120 mg vs. placebo: EQ-VAS (*p* = 0.0212), SWNS-PF (*p* = 0.0093), SI (*p* = 0.0368), and total score (*p* = 0.0395). MSQ with RBP-7000 vs. placebo (90 mg *p* = 0.0009, 120 mg *p* = 0.0006) RBP-7000 preferred over the previous medication (90 mg *p* = 0.0001, 120 mg *p* = 0.0619).
Litman et al., 2023 [81]	DB RCT + OLE	RLAI-OM 75 or 100 mg ISM, IM	Placebo	DB phase: Patients *n* = 433 OLE phase: Patients *n* = 215	DB phase: RLAI: 42.7 (10.89) placebo: 40.6 (11.23) OLE phase: RLAI 39.3 (10.84)	DB phase: patients with acute exacerbation of SZ OLE phase: patients who completed the DB phase, divided into three groups—stable, stabilized, unstable patients.	SZ	SWNS	Primary outcome: change in PSP and SWNS scale for both DB phase and OLE phase	DB phase: ↑ vs. placebo in single items of SWNS OLE phase: ↑ vs. baseline in single items of SWNS	DB phase: RLAI-ISM vs. placebo: SWNS-PF *p* < 0.006 at day 29. Any other single item change was not significant (*p* < 0.01) OLE phase: RLAI-ISM vs. baseline: SWNS-SI changes were significant at weeks 24 and 52 only for unstable patients (*p* < 0.01) SWNS total changes were significant at week 24 only for unstable patients (*p* < 0.01)
Mihajlović et al., 2011 [82]	Cross-sectional study	RLAI-TM 25–50 mg	HLAI-OM 50 mg	60	RLAI: 35.33 (7.02) HLAI: 50.97 (11.44)	SZ patients in treatment with HLAI 50 mg or RLAI 25–50 mg for longer than one year	SZ	SWLS, WHOQoL-BREF	Primary outcome: change in SFS, SWLS, WHO-QoL-Brief	↑ RLAI vs. ALO	WHOQoL-BREF: greater inclination to actively engage in everyday activities and leisure (*p* < 0.05) SWLS: higher satisfaction with life, living conditions, and self-image (*p* < 0.05)
Fe Bravo-Ortiz M et al., 2011 [83]	Observational longitudinal study	RLAI (dosage not specified)	Other AP: oral FGA, oral SGA, FGA-LAI	1865	<25 years: 193 (10.3%) 25–45 years: 1091 (58.5%) >45 years: 579 (31.0%) Not recorded: 2 (0.1%)	Patients initiating new antipsychotic treatment recruited by public mental health units and private clinics throughout Spain.	SZ, SZA, SZF	EQ-5D	Primary outcome: change in EQ-5D at months 3 and 6 Secondary outcome: change in PANSS total score and CGI-S at month 3 and 6	↑ vs. other LAI in improving EQ-5D	RIS LAI vs. other AP: higher percentage of patients (39.8%) with positive EQ-5D scores (in the past 3 months compared to other AP (*p* < 0.0018)

↑: increase/higher; ↔: similar; 5D: five dimensions; AP: antipsychotic; BP: Body Pain; CGI-S: clinical global inventory, severity; DB: double blind; EQ: EuroQoL; ER: Emotional Regulation; ESRS: Extrapyramidal Symptoms Rating Scale; FGA: first-generation antipsychotics; GAF: Global Assessment of Functioning; GH: General Health; HLAI: Haloperidol LAI; IM: intramuscular; ISM: in situ microparticles; LAI: Long-Acting Injectable; MH: Mental Health; MSQ: Medication Satisfaction Questionnaire; N/A: not applicable; NA: not available; OLE: open-label extension; OM: once a month; PANSS: Positive and Negative Syndrome Scale; PF: Physical Functioning; PP: paliperidone palmitate; QoL: quality of life; RCT: randomized controlled trial; RE: Emotional Role; RLAI: risperidone LAI; RP: Physical Role; SANS: Scale for the Assessment of Negative Symptoms; SAPS: Scale for the Assessment of Positive Symptoms; Sat: satisfaction; Sbc: subcutaneous; SC: Self Control; SF: Social Functioning; SF-36: Short Form-36 Health Survey; SFS: Social Functioning Scale; SGA: second-generation antipsychotics; SI: Social Integration; SWB: subjective well-being; SWLS: Satisfaction With Life Scale; SWN: Subjective Well-being under Neuroleptics; SWNS: Short form of the Subjective Well-being under Neuroleptics scale; SZ: schizophrenia; SZA: schizoaffective disorder; SZF: schizophreniform disorder; TM: twice a month; VAS: visual analog scale; WHOQoL-BREF: brief form of the World Health Organization Quality of Life scale.

**Table 4 jcm-12-06985-t004:** Summary of the included study about olanzapine LAI (OLAI).

Study	Study Design	Drugs	Comparator Group(s)	Sample	Age	Sample Characteristics	Diagnosis	Assessment of QoL/SWB/Sat	Outcome	Results	
										Change in QoL/SWB/Sat	Effect
Witte et al., 2012 [84]	Randomized, double-blind, placebo-controlled trial	OLAI-TM 210 mg; OLAI-TM 300 mg; OLAI-OM 405 mg	Placebo	404	210 mg: 39.8 (10.8); 300 mg: 41.5 (11.1); 405 mg: 39.5 (11.4); placebo: 42.6 (11.2)	BPRS ≥ 30 moderate-to-high level of symptom severity	SZ	SF-36	QLS, SF-36	↑ SF-36 vs. placebo	Combined group: better SF-36 MCS (*p* = 0.007), MH (*p* = 0.003), SF (*p* = 0.0149). MH significant (*p* < 0.005) in all OLAI doses, SC significant in 300 mg and 405 mg OLAI groups

↑: higher; BPRS: Brief Psychiatric Rating Scale; LAI: Long-Acting Injectable; MCS: mental component summary; MH: Mental Health subscale; OLAI: Olanzapine Long-Acting Injectable; OM: once a month; QLS: Heinrichs–Carpenter Quality of Life Scale; QoL: quality of life; SWB: subjective well-being; Sat: satisfaction; SF: Social Functioning; SF-36: Short Form-36 Health Survey; SZ: schizophrenia; TM: twice a month.

**Table 5 jcm-12-06985-t005:** Summary of the included studies on aripiprazole LAI once a month (AOM).

Study	Study Design	Drugs	Comparator Group(s)	Sample Size	Age	Sample Characteristics	Diagnosis	Assessment of QoL/SWB/Sat	Outcome	Results	
										Change in QoL/SWB/Sat	Effect
Schöttle et al., 2020 [85]	Observational longitudinal study	AOM 400 mg	N/A	242	43.1 (15.1)	Stable patients who switched their treatment to AOM after 9.7 (±22.3) months of oral treatment.	SZ	WHO-5 Index	Primary outcome: change from baseline at month 6 in BPRS, CGI, WHO-5, GAF	↑ vs. baseline in WHO-5 Index	At month 6, well-being significantly improved, with a 4.8-point increase on the WHO-5 index (*p* < 0.001). Initial progress was most prominent in the first 4 weeks
McEvoy et al., 2021 [86]	Post hoc analysis	AOM 400 mg	N/A	291	Age 18–70; median 38.6	Clinically stable patients	SZ	SF-36	Primary outcome: change in SF-36 mental component summary (MCS) and physical component summary (PCS) scores from baseline to 124 weeks	↑ of SF-36 vs. baseline	SF-36 MCS ↑ vs. baseline (*p* < 0.05); SF-36 PCS ↔ vs. baseline

↑: higher; ↔: similar; AOM: aripiprazole once a month; LAI: long acting antipsychotic; MCS: mental component summary; N/A: not applicable; PCS: physical component summary; QoL: quality of life; Sat: satisfaction; SF-36: Short Form-36 Health Survey; SWB: subjective well-being; WHO-5 Index: World Health Organization well-being index.

**Table 6 jcm-12-06985-t006:** Summary of the included studies on paliperidone palmitate LAI (PP).

Study	Study Design	Drugs	Comparator Group(s)	Sample Size	Age	Sample Characteristics	Diagnosis	Assessment of QoL/SWB/Sat	Outcome	Results	
										Change in QoL/SWB/Sat	Effect
Kwon et al., 2015 [87]	Randomized, open-label comparative study	PP1M	Oral SGA	134	34.3 (9.7)	Patients unsatisfied with the current treatment with atypical antipsychotics.	SZ	MSQ, TSQM	MSQ, TSQM, PANSS, PSP	↑ MSQ from baseline greater in PP1M; ↑ TSQM from baseline greater in PP1M	↑ MSQ (*p* < 0.0001) ↑ TSQM (*p* < 0.0001)
Takekita et al., 2016 [88]	Randomized, open-label, clinical trial	PP1M	RLAI-TM	30	PP1M; 43.5 (11.8) RLAI: 46.4 (10.4)	Non-acute phase; PANSS ≤ 120; RLAI for 2 months or longer	SZ, SZA	SWNS	Primary: BACS; Secondary: SWNS, PANSS, DIEPSS	↔ SWNS between groups	No between-group differences in SWNS
Sağlam Aykut, 2019 [89]	Observational longitudinal study	PP1M 100–150 mg	Oral SGA: quetiapine 600–1200 mg/d olanzapine 15–30 mg/d paliperidone 6–9 mg/d aripiprazole 15–30 mg/d risperidone 4–8 mg/d amisulpride 400–1200 mg/d clozapine 300–600 mg/d	84	PP1M: 36.91 (9.02) SGA oral: 37.24 (9.67)	Patients receiving treatment with either paliperidone palmitate or second-generation oral antipsychotics for at least six months	SZ	SF-36	Primary outcome: change at month 6 in PANSS, CGI, ESRS, UKU-SERS, SF-36, MMAS, SACTI	↑ vs. SGA oral in a single item of the SF-36	PAL LAI vs. oral SGA: SF-36, higher scores in the General Health subscale (*p* < 0.001). No significant differences were found in the other subscales
Di Lorenzo et al., 2022 [90]	Observational cohort study	PP1M, PP3M	HLAI	90	PP1M: 50.6 (13.4) PP3M: 51.4 (13.0) HLAI: 55.9 (11.7)	LAI therapy for at least 6 months.	SZ, SZA	WHOQOL-BREF	WHOQOL-BREF, GAF, CGI-S	↑ in WHOQOL-BREF vs. baseline within each group ↔ in WHOQOL-BREF between groups	↑ in WHOQOL-BREF vs. baseline for all groups (*p* < 0.006) No statistical difference between groups
Bozzatello et al., 2018 [28]	Open-label, randomized, controlled trial	PP1M	ER oral paliperidone	65	NA	Diagnosis of stable but symptomatic schizophrenia; previous unsuccessful treatment with an oral antipsychotic	SZ	TSQM, SWNS, SES	Primary: TSQM, SWNS, SES; Secondary: CGI-SCH, PSP	↑ TSMQ total and convenience subscale in the PP1M vs. oral paliperidone	↑ TSMQ total score (*p* = 0.001) ↑ TSMQ convenience (*p* = 0.037)
Fernández-Miranda et al., 2021 [91]	Observational longitudinal study	PP3M	PP1M	84	42.1 (7.6)	Patients with severe symptoms and impairment (GCI-S score ≥ 5) treated with PP3M after at least 2 years of stabilization with PP1M	SZ	TSQM, VAS-S	Primary outcome: change from baseline at month 24 in TSQM and VAS-S score	↑ TSQM and VAS-S in PP3M vs. PP1M	TSQM: *p* < 0.01 VAS-S: *p* < 0.001

↑ higher; ↔ similar; 1M: once a month; 3M: every three months; BACS: Brief Assessment of Cognition in Schizophrenia; CGI-S: Clinical Global Impression Severity; CGI-SCH: Clinical Global Impression of symptoms severity in Schizophrenia; DIEPSS: Drug-Induced Extrapyramidal Symptoms Scale; GAF: Global Assessment of Functioning; HLAI: haloperidol decanoate LAI; LAI: long-acting antipsychotic; MMAS: Morisky Medication Adherence Scale; MSQ: Medication Satisfaction Questionnaire; PANSS: Positive and Negative Syndrome Scale; PP: paliperidone palmitate; PSP: Personal and Social Performance scale; QoL: quality of life; RLAI: risperidone LAI; Sat: satisfaction; SF-36: Short Form-36 Health Survey; SGA: second-generation antipsychotics; SWNS: Short Form of the Subjective Well-being under Neuroleptics Scale; TM: twice a month; TSQM: Treatment Satisfaction Questionnaire for Medication; UKU-SERS: Ugvalg for Kliniske Undersgelser Side Effect Rating Scale; VAS-S: Visual Analog Scale for Satisfaction; WHOQoL-BREF: Brief form of the World Health Organization Quality of Life scale.

## Data Availability

Not applicable.

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
