# Peer review of "Second Generation Long-Acting Injectable Antipsychotics in Schizophrenia: The Patient’s Subjective Quality of Life, Well-Being, and Satisfaction"

_jcm, 2023, doi:10.3390/jcm12226985_

Round 1
Reviewer 1 Report
Comments and Suggestions for Authors
This manuscript entitled “Second generation long-acting injectable antipsychotics in schizophrenia: patient’s subjective quality of life, well-being, and satisfaction” points out a very important factor in delivering quality of care to patients with schizophrenia.
This is an important manuscript that brings together a plethora of information that highlights the need for more studies in this area, just as the authors recount.
This manuscript is well-written and their tables are very accessible. I have very little to add other than a few minor points.
It seems that the title of point 3.1 is missing.
I also believe that choosing to add the methods section in between their discussion and conclusions is unusual and even disruptive for the flow of the manuscript.
Reviewer 2 Report
Comments and Suggestions for Authors
The problem which is discussed in the manuscript provides rather unexpected but important form clinical point of view angle. The quality of the data is very good. The only remark – concentration of information is high and may be difficult to follow. Probably it worth to reorganize the in discussion section and to summarize separately rather objective quality of life data and rather subjective data on well-being.
Reviewer 3 Report
Comments and Suggestions for Authors
The present paper describes current evidence on the efficacy of second generation long-acting (SGA-LAI) antipsychotic regimen to improve subjective quality of life, well-being, and satisfaction in patients with schizophrenia. The studies evaluated include all possible molecules administered as SGA-LAI, proposing both recent and older studies The results described suggested that the evidence is still limited because of the lack of studies on this issue and because of several methodological issues including the study design, the samples of the studies, the number of studies for each drug and dosage employed, and the variegated assessment of the subjective quality of life, satisfaction, and well-being. Although the proposed results are limited by these issues, it is important that review studies such as this one highlight the role of SGA-LAI in improving the parameters of quality of life, well-being, and satisfaction and this must push clinicians to propose these treatments with greater emphasis.
The methodology is clear and adequate. The results are very exhaustive. The idea to summarize the assessment tools for quality of life, well-being, and satisfaction makes it much easier to read the results of the individual studies included. The discussion summarizes very well the strengths and weaknesses of the current literature evidence.
I have only a minor request. Altrought it is not necessary to realize a risk of bias analysis (RoB) for rapid reviews, as the authors reported to have analyzed this aspect, I suggest to add a figure describing RoB and to explain the results of ROB in the main text. Which tool has been applied to RoB Assessment?
Very minor points:
Line 86-89 grammar check required
Line 289: grammar check required
Line 400: this sentence is not linked to the context
Line 515: grammar check required
Lines 553-560: results of the presented study are not described in the main text but only in the table. Please describe them also in the text to be in line with the other cited studies.
Comments on the Quality of English Languageminor checks are required
Reviewer 4 Report
Comments and Suggestions for Authors
It is clear that the authors have spent considerable time in rewriting their manuscript for which they are to be commended. Abstract: write it in an impersonal way. Figures are very limited. Also, check all the acronyms carefully, since some of them do not have their description. For a more fluid reading of the article, add the methodology after the introduction. Since its a review article, all these are very important for the readers to understand better.
